# Paediatric Measles in Romania: A Comparative Clinical and Epidemiological Analysis of the 2017–2019 and 2023–2024 Epidemic Waves at a Tertiary Care Centre in Bucharest

**DOI:** 10.3390/v17060755

**Published:** 2025-05-26

**Authors:** Gheorghiță Jugulete, Mădălina Maria Merișescu, Bianca Borcos, Alexandra Nicoleta Totoianu, Anca Oana Dragomirescu

**Affiliations:** 1Department of Infectious Disease, Faculty of Dentistry, Carol Davila University of Medicine and Pharmacy, 050474 Bucharest, Romania; gheorghita.jugulete@umfcd.ro; 2National Institute for Infectious Diseases “Prof. Dr. Matei Bals”, No 1 Dr. Calistrat Grozovici Street, 021105 Bucharest, Romania; 3Department of Orthodontics and Dentofacial Orthopaedics, Faculty of Dentistry, “Carol Davila” University of Medicine and Pharmacy, 050474 Bucharest, Romania; anca.dragomirescu@umfcd.ro

**Keywords:** measles, children, clinical forms, complications, Romania

## Abstract

Measles remains a major public health issue, particularly among paediatric populations who are unvaccinated or lack of maternal antibody transfer. Although the majority of cases manifest with moderate clinical forms, certain patient categories are at risk for severe disease progression. This study aims to describe the clinical and epidemiological characteristics of paediatric measles cases hospitalized in the Paediatric Departments of the “Prof. Dr. Matei Balș” National Institute of Infectious Diseases, Bucharest, Romania during two distinct epidemic waves: 2017–2019 and 2023–2024. A retrospective analysis evaluated mortality rates, distribution by age and sex, as well as clinical disease patterns. The 2023–2024 measles epidemic was marked by a higher number of paediatric cases (3.114 vs. 1.068), a lower mortality rate (0.32% vs. 3.74%), a shift towards older age groups, and a greater frequency of complications—particularly gastrointestinal, haematological, and ophthalmological—compared to the 2017–2019 wave. The findings underscore the urgent need for strengthened vaccination programs and targeted public health interventions, particularly among vulnerable groups and patients at risk of developing severe forms of the disease. Owing to a sustained decline in measles vaccination coverage among the paediatric population, Romania has experienced two major measles outbreaks within the past decade, interrupted by the COVID-19 pandemic. This study draws attention to the increasing incidence of measles in older children, suggesting a cumulative effect of reduced immunization rates over time.

## 1. Introduction

Over the past decade, Romania has experienced two major measles epidemic waves, occurring in 2017–2019 and 2023–2024, with a notable intermission during the COVID-19 pandemic (2019–2022). These outbreaks have developed in the context of a declining vaccination rate [1,2,3,4,5].

The World Health Organization (WHO) has set an ambitious goal to eliminate measles in all regions by 2030, emphasizing high vaccination coverage (≥95% with two doses of measles-containing vaccine), robust surveillance systems, and rapid outbreak response as essential pillars of this strategy [6]. In this context, our study contributes valuable epidemiological data from a high-burden EU country, documenting patterns of clinical severity, and gaps in vaccine coverage during two major outbreaks over the past few years—serving as a meaningful stop along the WHO’s global roadmap towards measles elimination.

Measles, an acute childhood disease with an extremely high degree of contagiousness, is characterized in its prodromal phase by fever, cough, and coryza, followed by the appearance of oral enanthem (Koplik spots) and a descending maculopapular exanthem. The acute phase of the disease is often accompanied by complications, some of which may be severe, leading to long-term sequelae or fatal outcomes. The severity of these complications is largely determined by the host’s immunocompromised status, which is further intensified by virus-mediated immune suppression and the direct cytopathic effects. Consequently, measles can lead to respiratory complications (giant cell pneumonia, bronchopneumonia, pleural effusion, capillary bronchiolitis, acute respiratory failure); digestive complications (dehydration syndrome, hepatitis, pancreatitis, stomatitis); otolaryngological complications (suppurative otitis, croup laryngitis); ocular complications (keratoconjunctivitis); and neurological complications (meningitis, encephalitis, myelitis). The most severe among these are encephalitic manifestations, including subacute encephalitis, inclusion-body encephalitis, and subacute sclerosing panencephalitis (SSPE). These can occur both during the acute phase of the disease and, more frequently, in the post-eruptive stage or at a later time through an autoimmune mechanism, often progressing with severe outcomes and, in the case of SSPE, even resulting in death [7,8,9].

Measles spreads primarily through direct contact with respiratory droplets and, less commonly, through airborne transmission. The incubation period typically ranges from 10 to 14 days after exposure. Individuals infected with measles are contagious from approximately four days before to four days after the appearance of the rash. The disease is extremely contagious, with a basic reproductive number (R0) estimated between 12 and 18, meaning that one infected person can transmit the virus to 12 to 18 others in a fully susceptible population. Around 90% of people who come into contact with the virus and lack immunity will develop symptoms [10].

The clinical forms of measles range from mild or mitigated to moderate or severe, depending on the host’s immune status and pre-existing antibody titre. The main risk factors for measles include lack of vaccination, malnutrition, pre-existing chronic respiratory diseases, young age, and immunosuppression. Unvaccinated individuals, particularly children, remain the most susceptible to infection and severe complications, especially in countries with suboptimal immunization coverage [11,12]. Malnutrition, particularly vitamin A deficiency, has been consistently associated with increased disease severity, prolonged recovery, and higher mortality [12,13,14]. Children with chronic respiratory conditions, such as asthma, may experience more severe pulmonary complications, including pneumonia [12,13]. Severe cases occurring in immunocompromised individuals can have an unfavourable course, leading to multisystem involvement, multiple organ failure, and even death [9,15,16].

The aim of this study was to compare the clinical and epidemiological characteristics of paediatric measles cases hospitalized in our centre during two major epidemic waves in Romania in order to identify changes in disease presentation and outcomes over time.

## 2. Materials and Methods

We conducted a retrospective descriptive study of measles cases hospitalized in the Paediatric Departments of the “Prof. Dr. Matei Balș” National Institute of Infectious Diseases, a tertiary hospital in Bucharest, capital of Romania, Southeastern Europe during the periods 2017–2019 and 2023–2024.

The study included paediatric patients aged 0 to 18 years, who were categorized into two cohorts according to their period of hospitalization: Group 1—1.068 cases (2017–2019) and Group 2—3.114 cases (2023–2024). The diagnosis of measles was established based on a combination of epidemiological, clinical, laboratory, and paraclinical criteria. Clinical diagnosis is based on the presence of fever and rash, along with at least one of the following four signs: cough, coryza, conjunctivitis, or Koplik spots. Laboratory investigations included nasopharyngeal swabs for pathogen identification using polymerase chain reaction (PCR) testing and blood samples for serological analysis, with a positive measles-specific IgM result indicating infection. Criteria for hospitalization include moderate to severe or complicated forms (e.g., pneumonia, encephalitis, laryngitis, acute respiratory failure, etc.); young age (<1 year); poor general condition; dehydration or inability to tolerate oral fluids; children with comorbidities (e.g., malnutrition, immunodeficiency); and secondary bacterial complications.

Demographic data (age and sex) and clinical characteristics were extracted from patients’ medical records.

The data from the periods 2017–2019 and 2023–2024 were obtained from the hospital’s statistical program, Info World. We selected the period and the diagnosis of measles infection, both as primary and secondary diagnoses, and then counted the patients. Data collection was performed using an MS Excel database, ensuring the confidentiality of the study participants by protecting their personal data. Upon admission, all legal caregivers signed a consent form for the publication and the use of patient data for the study. The analysis was conducted in accordance with the Declaration of Helsinki and was approved by the Ethics Committee of the National Institute of Infectious Diseases “Prof. Dr. Matei Balș”.

Statistical analysis was performed using Fisher’s exact test for comparison of proportions and Chi-square test. Categorical variables are presented as absolute values and percentages. Data analysis was conducted using R software (version 4.4.2), developed by the R Core Team (2024), with reference to the R Foundation for Statistical Computing, Vienna, Austria (URL: https://www.R-project.org).

## 3. Results

The study showed two significant waves of measles outbreaks. The first epidemic wave recorded between June 2017 and October 2019 (Group 1: 2017–2019), and a total of 1068 paediatric measles cases were documented, with a mortality rate of 3.74%. In contrast, the most recent epidemic wave recorded between August 2023 and December 2024 (Group 2: 2023–2024) saw a notable increase in the number of paediatric cases, with 3114 reported and a single fatality, corresponding to a mortality rate of 0.32% (Table 1).

In terms of sex distribution among children, no statistically significant differences were observed between the two groups (*p* = 0.508). We observed that the number of male children predominated compared to the number of female children (*p* < 0.001). The proportion of boys was 55.3% in the first group and 53.5% in the second group (Table 1).

An age group analysis of hospitalized paediatric patients showed that during the 2017–2019 epidemic, a higher number of cases were reported among infants aged 0 to 1 year compared to the 2023–2024 epidemic (*p* < 0.00001). The number of cases among preschool-aged children (1–6 years) remained comparable across both epidemic periods. Notably, a relative increase in the proportion of school-aged children, 6–14 years (*p* = 0.0001) and adolescents (*p* < 0.00001) was observed during the 2023–2024 wave (Table 1).

Regarding vaccination status, in our study, the majority of measles cases occurred in unvaccinated individuals. In the last epidemic, 98.2% of patients were unvaccinated compared to the 2017–2019 epidemic wave, when 96.7% of cases were unvaccinated (*p* = 0.007). This higher proportion of unvaccinated patients in Group 2 is also reflected among those who received only one dose of vaccine, although they had reached the booster age (Group 1—2.7%, Group 2—1.6%, *p* = 0.026) (Table 1).

Figure 1 and Figure 2 demonstrates a significant rise in the number of paediatric measles cases during the most recent epidemic wave. This sharp increase may reflect the greater impact and rapid spread of the measles virus in a population with low vaccination coverage, emphasizing the vulnerability of unvaccinated groups.

Figure 3 illustrates that moderate clinical forms of measles predominated among hospitalized children in both epidemics. Compared to 2017–2019, moderate forms were more frequent (*p* < 0.001), while in 2023–2024, mild (*p* < 0.001) or severe (*p* = 0.036) forms were more frequent. Severe measles cases represented patients who had associated acute respiratory failure, SIRS, sepsis, severe dehydration, and neurological complications. In the previous epidemic, despite a lower percentage of severe measles cases, there were more critical cases, with four deaths 3.74%). The children who died were of very young age, three of them being infants under the age of one, and one child aged two.

The complications associated with paediatric measles in the last epidemic are more frequent and severe compared to the previous wave. Complications were recorded in the majority of hospitalized paediatric cases (92.3%), with an increase in severity (Table 2).

Respiratory and systemic complications were identified with an approximately equal proportion in both groups. The most frequent respiratory complications were acute respiratory failure, pneumonia, and bronchopneumonia and represented a large percentage of the total complications. Of these, eight cases required intubation and mechanical ventilation. Systemic complications were represented by sepsis, myositis, and prolonged febrile syndrome.

Gastrointestinal (acute diarrhoea, hepatitis, dehydration syndrome, pancreatitis, stomatitis); haematological (leukopenia, anaemia, thrombocytopenia); and ophthalmological (conjunctivitis and keratoconjunctivitis) complications were more frequent in 2023–2024 (*p* < 0.00001). Neurological complications (seizures and encephalitic reaction) were more frequent in 2017–2019 (Table 2).

## 4. Discussion

The comparison between the two epidemic waves highlights a significant increase in the number of paediatric measles cases during the 2023–2024 outbreak. This surge may be attributed to the declining vaccination coverage, as nearly all affected children were unvaccinated. Despite the higher case count, the mortality rate was lower compared to the 2017–2019 epidemic. This suggests an improvement in early diagnosis and supportive care.

Our study found that the last epidemic wave (2023–2024) particularly affected individuals under 1 year of age and those older than 14 years old. This shift may reflect vaccine hesitancy or missed immunization opportunities in older children. These findings point to a more complex clinical presentation and a greater burden on healthcare services.

Complication rates were significantly elevated during the 2023–2024 wave. Respiratory complications were most common, with several cases requiring intensive care. Gastrointestinal and haematological complications were also frequently observed in the last wave. Although neurological and systemic complications were less prevalent, they posed serious threats to patient outcomes. The increased frequency and severity of complications highlight the critical need for high vaccination coverage and the development of standardized treatment protocols in paediatric care settings.

### 4.1. The National and International Context of the Measles Epidemic

During both the 2017–2019 and 2023–2024 periods, the number of measles cases remained high across Europe, influenced by increased population mobility through travel and migration. An alarming rise in measles incidence was observed in the post-pandemic period, driven by several major factors: the isolation measures imposed during the COVID-19 pandemic, disruption of routine immunization programs, and growing parental hesitancy and scepticism regarding vaccination [17,18,19,20,21].

In Romania, 31,886 measles cases were recorded during the two years of the 2023–2024 epidemic, compared to 19,520 cases during the 2015–2019 period. In 2017, Romania reported the highest number of measles cases in Europe (5608), followed by Italy (5098) [22]. Moreover, in 2024, the number of measles cases across Europe doubled compared to 2023, reaching the highest level in the past 25 years. According to UNICEF and the World Health Organization (WHO), Romania reported the highest number of measles cases in the European Region for 2024 (31,886 number of cases), followed by Kazakhstan (28,147 number of cases) [20,22].

In Romania, the incidence of measles has exhibited significant fluctuations over recent years, reflecting the dynamics of epidemiological trends and vaccination coverage. In 2018, the national incidence was recorded at 42.77 cases per 100,000 population, a decrease from 46.2 cases per 100,000 in 2017. By 2019, the incidence further declined to 21.26 cases per 100,000. However, in 2023, the incidence was 19.3 cases per 100,000. Preliminary data for 2024 indicate a significant increase in the number of measles cases; however, the exact national incidence rate is not yet available [20]

In Romania, measles surveillance is conducted using a national epidemiological system coordinated by the National Institute of Public Health (INSP), in alignment with WHO and ECDC guidelines. The surveillance strategy mandates the immediate reporting of all suspected cases by healthcare providers, followed by epidemiological investigation and laboratory confirmation through serology (IgM ELISA) or molecular testing (RT-PCR). Although data collection is compulsory, the system’s effectiveness can be hindered by underreporting, delayed notifications, and limited access to diagnostic testing in certain areas. Despite these challenges, the surveillance system has proven effective in detecting outbreaks and monitoring epidemiological trends [20].

An important point to highlight is that, at the European level, the majority of measles cases were reported in children under five years of age. In contrast, Romania registered a significant number of cases also among older age groups, a phenomenon explained by the lack of adequate immunization during early childhood [11,20,22]. Below, in Table 3, we present the MMR vaccination coverage trends in Romania, highlighting a concerning decline in recent years, with coverage levels falling well below the desired target of over 95% [20].

As of early April 2025, the United States has reported over 700 measles cases, making it the highest annual count since 2019. The majority of these cases—more than 500—have been recorded in Texas, which remains the most heavily affected state in the current outbreak. This has predominantly impacted children and adolescents, most of whom were either unvaccinated or had an unclear immunization status with two fatalities declared [23,24].

### 4.2. Epidemiological Impact on Healthcare Systems

Across Europe, over half of the reported measles cases required hospitalization, placing considerable strain on public health systems, particularly paediatric departments. Due to its high contagion rate, measles has posed serious challenges in managing healthcare resources and hospital capacity [21,22,24]. Historically, genotype D4 has been the endemic strain of the measles virus in Romania. Since 2016, however, additional genotypes—most notably B3 and D8—have been identified during measles outbreaks, suggesting repeated importations of the virus from other regions. Between March and August 2023, new variants of the D8 genotype were detected in the western part of the country, indicating a genetic diversification of the circulating strains in Romania. Phylogenetic analyses revealed two distinct clusters of the D8 genotype, differing from those identified in previous outbreaks between 2011 and 2019, which implies multiple introductions of the virus into the country [20,23,24].

In our case, the hospitalization of moderate to severe cases due to complications has overwhelmed paediatric wards, complicated case allocation due to the high contagiousness of the virus, and—when overlapping with other epidemics such as the 2024 influenza outbreak or pertussis—has further strained hospital resources and complicated case management [21,25]. In addition to the clinical burden, the outbreak has led to significant direct costs, including increased expenditures for hospitalization, intensive care, diagnostics, and supportive therapies. Indirect costs have also risen, caused by caregiver absenteeism, school closures, and long-term health impacts in affected children [26,27,28].

### 4.3. Vaccination Policies and Prevention Strategies

In our country, the current vaccination policy is regulated through the National Vaccination Program (NVP), coordinated by the Ministry of Health and implemented through the network of family doctors. This policy aligns with the National Vaccination Strategy 2023–2030, which aims to increase vaccination coverage and reduce the incidence of vaccine-preventable diseases. Vaccination is not legally mandatory; the recommended vaccines are offered free of charge and are strongly encouraged as part of public health efforts [27,29].

In Europe, MMR vaccination is recommended in 24 countries and mandatory in 17, including Albania, Bosnia and Herzegovina, Bulgaria, Czech Republic, France, Hungary, Italy, Malta, Moldova, Montenegro, North Macedonia, Russia, Serbia, Slovakia, Slovenia, and Ukraine. In Germany, measles vaccination is mandatory, while immunization against mumps and rubella remains optional [19,29].

The World Health Organization (WHO) and UNICEF are working with national governments to strengthen immunization programs, engage communities in outbreak prevention, and train healthcare personnel. These efforts aim to achieve optimal vaccination coverage to prevent a resurgence of measles, reduce the occurrence of severe disease, and mitigate the impact on public health systems [29].

In Romania, an effective measles vaccination strategy must extend beyond routine immunization to address the needs of high-risk and vulnerable populations. Targeted public health interventions is essential in some communities, including socio-economically disadvantaged groups, migrants, and institutionalized children, where immunization rates are often suboptimal due to limited healthcare access and low health literacy. Public health interventions should prioritize transparent, evidence-based communication, engage trusted local figures, and actively counter misinformation, especially in online spaces. Training primary care providers in vaccine counselling further supports this effort, fostering trust between caregivers and the healthcare system. Robust electronic immunization registries can enhance real-time coverage tracking, identify unvaccinated individuals, and guide catch-up campaigns. A relevant example comes from Denmark, where a proactive electronic reminder system notifies parents to schedule vaccinations as part of the national immunization program. Such integrated, proactive strategies are vital to closing immunity gaps and preventing future measles outbreaks, particularly in the aftermath of declining vaccine coverage during and after the COVID-19 pandemic [26,28,29].

The main limitations of this study include its retrospective, single-centre design, and the fact that only hospitalized cases were analysed. This may lead to an overestimation of the severity of measles infections, as the data do not capture the full clinical spectrum of the disease, particularly milder cases managed in outpatient care or those isolated at home without requiring hospitalization.

## 5. Conclusions

This study highlights the persistent severity of measles and underscores key clinical and epidemiological findings observed during the 2023–2024 epidemic in Romania. Our study highlights an increasing number of cases in older children, reflecting a decrease in measles vaccination coverage. Notably, there was a marked increase in complication rates compared to previous outbreaks, with respiratory manifestations being the most prevalent and often requiring intensive care. Gastrointestinal and haematological complications were also frequently documented, while neurological and systemic complications, though less common, remained clinically significant due to their impact on patient outcomes.

Despite a higher overall case count, the mortality rate was lower than during the 2017–2019 epidemic, suggesting improvements in early diagnosis and supportive management. The increased burden of moderate to severe cases placed substantial pressure on paediatric wards, particularly during periods of overlap with other epidemics such as influenza and pertussis. These findings illustrate the strain placed on healthcare resources and highlight the consequences of declining vaccination coverage.

The study further emphasizes the necessity of both specific interventions—such as the timely administration of measles immunoglobulin to high-risk children—and non-specific preventive strategies aimed at reducing transmission. Overall, our data reinforce the urgent need to address declining vaccination rates and implement effective public health strategies to control measles transmission.

## Figures and Tables

**Figure 1 viruses-17-00755-f001:**
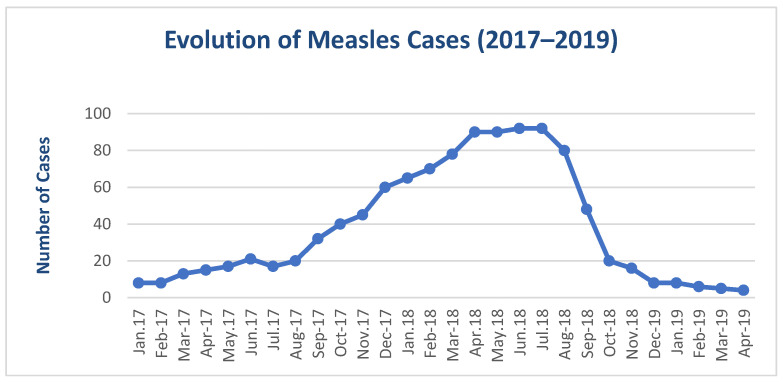
Evolution of paediatric measles cases during the 2017–2019 period.

**Figure 2 viruses-17-00755-f002:**
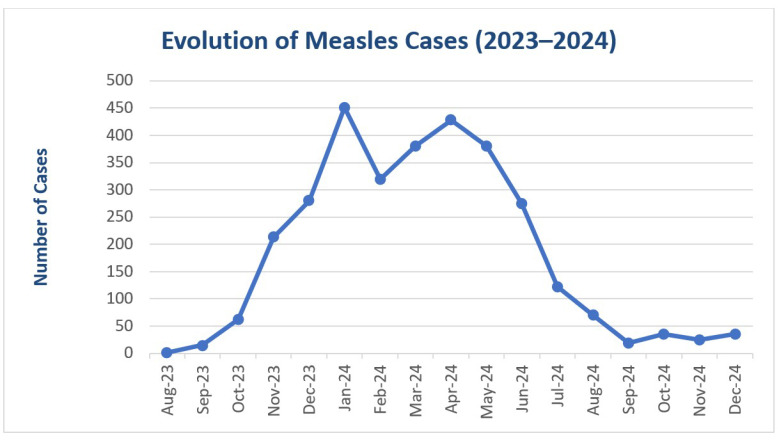
Evolution of paediatric measles cases during the 2023–2024 period.

**Figure 3 viruses-17-00755-f003:**
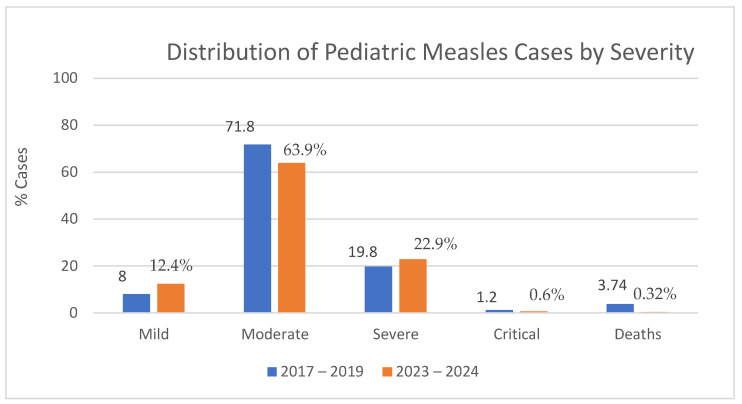
Distribution of hospitalized paediatric measles cases by clinical severity.

**Table 1 viruses-17-00755-t001:** Comparative characteristics in epidemic waves of measlesignificant (*p* < 0.05).

Characteristics	Group 1 (2017–2019)	Group 2 (2023–2024)	*p*-Value
**Age group**			
0–1 years old	351 (32.9%)	646 (2.7%)	*p* < 0.001
1–6 years old	484 (45.3%)	1504 (48.3%)	*p* = 0.087
6–14 years old	198 (18.5%)	755 (24.2%)	*p* < 0.001
Over 14 years old	34 (3.2%)	209 (6.7%)	*p* < 0.001
Gender			*p* = 0.508
Male sex	591 (55.3%)	1666 (53.5%)	
Female sex	477 (44.7%)	1448 (46.5%)	
Mortality rate	4 (3.74%)	1 (0.32%)	*p* = 0.016
Vaccination status			
Unvaccinated	1033 (96.7%)	3058 (98.2%)	*p* = 0.007
Incompletely vaccinated for age (one dose)	29 (2.7%)	50 (1.6%)	*p* = 0.026
Fully vaccinated	6 (0.6%)	6 (0.2%)	*p* = 0.157

**Table 2 viruses-17-00755-t002:** Measles complications in children observed during the 2017–2019 and 2023–2024 outbreaks. Significant (*p* < 0.05).

Complications	Group 1	Group 2	*p*-Value
Haematological	409 (38.3%)	1454 (46.7%)	*p* < 0.001
Respiratory	838 (78.5%)	2491 (80%)	*p* = 0.305
ENT	154 (14.4%)	368 (12.4%)	*p* = 0.030
Digestive	880 (82.4%)	2771 (89%)	*p* < 0.001
Ophthalmological	317 (29.7%)	1162 (37.3%)	*p* < 0.001
Neurological	26 (2.4%)	53 (1.7%)	*p* < 0.001
Systemic	19 (1.8%)	65 (2.1%)	*p* = 0.622

**Table 3 viruses-17-00755-t003:** MMR vaccination coverage (%) in Romania; source: https://insp.gov.ro/.

Country	Vaccine	Data Source	2024	2023	2022	2021	2020	2019	2018	2017
Romania	MMR, 1st dose	* OFFICIAL	65.5%	78%	83.4%	86.2%	87.3%	89.5%	89.6%	86.5%
		** WUENIC	78%	78%	83%	86%	87%	90%	90%	86%
	MMR, 2nd dose	OFFICIAL	68.1%	62.1%	71.4%	74.6%	75.1%	75.8%	80.9%	74.7%
		WUENIC	62%	62%	71%	75%	75%	76%	81%	75%

* OFFICIAL—report of National Institute of Public Health Romania; ** WUENIC—WHO/UNICEF Estimates of National Immunization Coverage.

## Data Availability

The datasets generated and analysed during the current study are available from the corresponding author upon request.

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
