# Peer review of "Paediatric Measles in Romania: A Comparative Clinical and Epidemiological Analysis of the 2017–2019 and 2023–2024 Epidemic Waves at a Tertiary Care Centre in Bucharest"

_viruses, 2025, doi:10.3390/v17060755_

Round 1
Reviewer 1 Report
Comments and Suggestions for Authors
Overall, the scientific article is well-structured and presents a relevant study on an important public health issue. Here's a breakdown of its strengths and areas where it could be improved.
Abstract
The abstract is well-structured but consider adding specific data points to highlight the key findings. For example, "The 2023-2024 measles epidemic was marked by a higher number of paediatric cases (3,114 vs. 1,068), a lower mortality rate (0.32% vs. 3.74%), a shift towards older age groups, and a greater frequency of complications..."
Add a sentence or two about the public health implications or the importance of the findings.
Introduction
The introduction provides a good overview of measles and the context of the study.
I would recommend expanding the introduction to include details on the modes of transmission and key risk factors for measles. This addition would provide a more comprehensive context for the reader and strengthen the rationale for the study. In addition, consider adding a sentence about the specific objectives of the study to provide a clearer focus for the reader.
Materials and Methods
The Materials and Methods section is clear and concise. However, it might be helpful to provide more detail on the clinical and paraclinical criteria used for measles diagnosis. This would enhance the reproducibility of the study.
Regarding the statistical analysis, it is not clear why the use of the exact Fisher test should be better justified. This test finds indications in the case of small numbers, therefore its use is justified only in the case of mortality, not for other comparisons for which the chi-square test would be more suitable.
Results
In the Results section, it would be clearer to present the data in a more organized manner. For example, when presenting p-values, ensure consistency in the number of decimal places; report the exact value of p even when greater than 0.05; always use the point as decimal separator; align the headers of the two tables; in Table 1 the heading "Demographic features" should be replaced with gender.
The presentation of data in the text should complement the tables and graphs. Avoid redundancy but emphasize key findings. Clearly label Graph 1, Graph 2, and Graph 3. The labeling in the text is a bit confusing.
Discussion
The Discussion section effectively interprets the results and relates them to the existing literature.
The study clearly shows a rise from 1,068 cases in 2017-2019 to 3,114 cases in 2023-2024. However, when discussing the reasons for this increase, and especially when linking it to vaccination coverage, relying solely on absolute numbers is problematic. To strengthen the discussion, I recommend including specific data on vaccination coverage rates in Romania during the 2017-2019 and 2023-2024 periods. This would provide more robust evidence to support the claims regarding the impact of declining vaccination on the increased number of measles cases. Additionally, consider presenting incidence rates (cases per 100,000 children) to account for potential changes in the population size.
The discussion of the international context is relevant and important. Consider expanding on the reasons for vaccine hesitancy and skepticism, as this is a critical factor driving measles outbreaks. Strengthen the discussion by providing more context on the implications of the findings for public health policy and clinical practice. For example, discuss the need for targeted interventions to improve vaccination coverage in specific age groups or the development of clinical guidelines for managing severe measles cases.
Finally, there are no references to the limitations of the study, for example its single-centre design.
Conclusions
The conclusions are clear and concise.
Reinforce the main findings and their implications. For example, "The increasing number of measles cases in older children and the higher frequency of severe complications in the 2023-2024 epidemic highlight the urgent need to address declining vaccination rates and implement effective public health strategies to control measles transmission."
Finally, the sentence "The findings underscore the importance of both non-specific preventive measures, such as avoiding crowded areas, maintaining personal and community hygiene, and isolating individuals with confirmed measles, and specific preventive strategies, including the administration of anti-measles immunoglobulin to at-risk children and increasing vaccine coverage" is partially supportable by the results of the study, but requires some nuance. The statement is generally reasonable from a public health perspective. However, to be strictly accurate in the context of this specific study, the authors should emphasize the strong evidence for increasing vaccine coverage. The other preventive measures, while valid, are not as directly supported by the data presented.
Overall, the article is well-written and provides valuable insights into the epidemiology and clinical characteristics of measles in Romania. By addressing the suggestions above and performing a general check on text formatting, you can further enhance the clarity, rigor, and impact of your work.
Author Response
Response to Reviewer 1:
Thank you for your valuable comments and suggestions. In response:
- A new author, who was not included in the initial version, has been added to the manuscript. Consequently, the order of authors has changed.
- The graphs have been revised and renamed for greater clarity and consistency.
- Additionally, we have included a new table presenting vaccination rates during the analyzed periods, to improve data interpretation and support the discussion.
We appreciate your feedback, which has contributed to improving the quality of the manuscript.
Overall, the scientific article is well-structured and presents a relevant study on an important public health issue. Here's a breakdown of its strengths and areas where it could be improved.
Thank you for your valuable feedback and kind support.
Abstract
The abstract is well-structured but consider adding specific data points to highlight the key findings. For example, "The 2023-2024 measles epidemic was marked by a higher number of paediatric cases (3,114 vs. 1,068), a lower mortality rate (0.32% vs. 3.74%), a shift towards older age groups, and a greater frequency of complications..."
We added specific data points The 2023–2024 measles epidemic was marked by a higher number of paediatric cases (3.114 vs 1068), a lower mortality rate (0.32 ‰ vs 3.74 ‰),
Add a sentence or two about the public health implications or the importance of the findings.
We added the idea: The findings underscore the urgent need for strengthened vaccination programs and targeted public health interventions, particularly among vulnerable groups and patients at risk of developing severe forms of the disease
Introduction
The introduction provides a good overview of measles and the context of the study.
I would recommend expanding the introduction to include details on the modes of transmission and key risk factors for measles. This addition would provide a more comprehensive context for the reader and strengthen the rationale for the study. In addition, consider adding a sentence about the specific objectives of the study to provide a clearer focus for the reader.
We added three ideas:
Measles spreads primarily through direct contact with respiratory droplets and, less commonly, through airborne transmission. The incubation period typically ranges from 10 to 14 days after exposure. Individuals infected with measles are contagious from approximately four days before to four days after the appearance of the rash. The disease is extremely contagious, with a basic reproductive number (R0) estimated between 12 and 18, meaning that one infected person can transmit the virus to 12 to 18 others in a fully susceptible population. Around 90% of people who come into contact with the virus and lack immunity will develop symptoms.
The main risk factors for measles include lack of vaccination, malnutrition, pre-existing chronic respiratory diseases, young age, and immunosuppression. Unvaccinated individuals, particularly children, remain the most susceptible to infection and severe complications, especially in countries with suboptimal immunization coverage [1,2]. Malnutrition, and especially vitamin A deficiency, has been consistently associated with increased disease severity, prolonged recovery, and higher mortality [2–4]. Children with chronic respiratory conditions, such as asthma, may experience more severe pulmonary complications, including pneumonia [2,5].
The aim of this study was to compare the clinical and epidemiological characteristics of paediatric measles cases hospitalized in our center during two major epidemic waves in Romania (2017–2019 and 2023–2024), in order to identify changes in disease presentation and outcomes over time.
Materials and Methods
The Materials and Methods section is clear and concise. However, it might be helpful to provide more detail on the clinical and paraclinical criteria used for measles diagnosis. This would enhance the reproducibility of the study.
Regarding the statistical analysis, it is not clear why the use of the exact Fisher test should be better justified. This test finds indications in the case of small numbers, therefore its use is justified only in the case of mortality, not for other comparisons for which the chi-square test would be more suitable.
We completed with clinical and paraclinical criteria. Also we added criteria for hospitalization.
Clinical diagnosis is based on the presence of fever and rash, along with at least one of the following three signs: cough, coryza, or conjunctivitis. Laboratory investigations included nasopharyngeal swabs for pathogen identification using polymerase chain reaction (PCR) testing and blood samples for serological analysis — with a positive measles-specific IgM result. Criteria for hospitalization include: moderate to severe or complicated forms (e.g., pneumonia, encephalitis, laryngitis, acute respiratory failure, etc.); young age (<1 year); poor general condition; dehydration or inability to tolerate oral fluids; children with comorbidities (e.g., malnutrition, immunodeficiency); and secondary bacterial complications.
We used Fisher's exact test to compare mortality and the rate of fully vaccinated children. For the other analyses we used the Chi-square test.
We mentioned also in the manuscript Statistical analysis was performed using Fisher’s exact test for comparison of proportions and Chi-square test.
Results
In the Results section, it would be clearer to present the data in a more organized manner. For example, when presenting p-values, ensure consistency in the number of decimal places; report the exact value of p even when greater than 0.05; always use the point as decimal separator; align the headers of the two tables; in Table 1 the heading "Demographic features" should be replaced with gender.
We rewrote the tables and added p-values even when p-values > 0.05. I changed the subtitle of the table ,,demographic data’’ to ,,gender’’
The presentation of data in the text should complement the tables and graphs. Avoid redundancy but emphasize key findings. Clearly label Graph 1, Graph 2, and Graph 3. The labeling in the text is a bit confusing.
The graphs have been revised and renamed.
Discussion
The Discussion section effectively interprets the results and relates them to the existing literature.
The study clearly shows a rise from 1,068 cases in 2017-2019 to 3,114 cases in 2023-2024. However, when discussing the reasons for this increase, and especially when linking it to vaccination coverage, relying solely on absolute numbers is problematic. To strengthen the discussion, I recommend including specific data on vaccination coverage rates in Romania during the 2017-2019 and 2023-2024 periods. This would provide more robust evidence to support the claims regarding the impact of declining vaccination on the increased number of measles cases. Additionally, consider presenting incidence rates (cases per 100,000 children) to account for potential changes in the population size.
We added an idea about vaccination coverage rates in Romania in both periods. We presented incidence rate per 100.000 population, in Romania only in 2017-2019 period; for 2023-2024 we don’t have specific data.
Additionally, a new table with vaccination rates from the analyzed periods has been created to provide a clearer understanding of the data.
The discussion of the international context is relevant and important. Consider expanding on the reasons for vaccine hesitancy and skepticism, as this is a critical factor driving measles outbreaks. Strengthen the discussion by providing more context on the implications of the findings for public health policy and clinical practice. For example, discuss the need for targeted interventions to improve vaccination coverage in specific age groups or the development of clinical guidelines for managing severe measles cases.
Finally, there are no references to the limitations of the study, for example its single-centre design.
We added: In Romania, an effective measles vaccination strategy must extend beyond routine immunization to address the needs of high-risk and vulnerable populations.
We completed the discutions with vaccination coverage in Romania over the last years. Also we tried to find the incidence of cases and we noted the incidente per 100.000 population between 2017-2019
We added the limitations of the study: The main limitations of this study include its retrospective, single-center design, and the fact that only hospitalized cases were analyzed.
Conclusions
The conclusions are clear and concise.
Reinforce the main findings and their implications. For example, "The increasing number of measles cases in older children and the higher frequency of severe complications in the 2023-2024 epidemic highlight the urgent need to address declining vaccination rates and implement effective public health strategies to control measles transmission."
Finally, the sentence "The findings underscore the importance of both non-specific preventive measures, such as avoiding crowded areas, maintaining personal and community hygiene, and isolating individuals with confirmed measles, and specific preventive strategies, including the administration of anti-measles immunoglobulin to at-risk children and increasing vaccine coverage" is partially supportable by the results of the study, but requires some nuance. The statement is generally reasonable from a public health perspective. However, to be strictly accurate in the context of this specific study, the authors should emphasize the strong evidence for increasing vaccine coverage. The other preventive measures, while valid, are not as directly supported by the data presented.
We changed the conclusions in this way
This study highlights the persistent severity of measles and underscores key clinical and epidemiological findings observed during the 2023–2024 epidemic in Romania. Notably, there was a marked increase in complication rates compared to previous outbreaks, with respiratory manifestations being the most prevalent and often requiring intensive care. Our study highlights an increasing number of cases in older children, reflecting a decrease in measles vaccination coverage. Gastrointestinal and haematological complications were also frequently documented, while neurological and systemic complications, though less common, remained clinically significant due to their impact on patient outcomes.
Despite a higher overall case count, the mortality rate was lower than during the 2017–2019 epidemic, suggesting improvements in early diagnosis and supportive management. The increased burden of moderate to severe cases placed substantial pressure on paediatric wards, particularly during periods of overlap with other epidemics such as influenza and pertussis. These findings illustrate the strain placed on healthcare resources and highlight the consequences of declining vaccination coverage.
The study further emphasizes the necessity of both specific interventions—such as the timely administration of measles immunoglobulin to high-risk children—and non-specific preventive strategies aimed at reducing transmission. Overall, our data reinforce the urgent need to address declining vaccination rates and implement effective public health strategies to control measles transmission.
Overall, the article is well-written and provides valuable insights into the epidemiology and clinical characteristics of measles in Romania. By addressing the suggestions above and performing a general check on text formatting, you can further enhance the clarity, rigor, and impact of your work.
Thank you for your support!

Reviewer 2 Report
Comments and Suggestions for Authors
I answered 'yes' to the question regarding inappropriate self-citations because 7 out of the 20 references cited were authored by the same individual as the current paper.

Author Response
Response to Reviewer 2:
Thank you for your valuable comments and suggestions. In response:
- A new author, who was not included in the initial version, has been added to the manuscript. Consequently, the order of authors has changed.
- The graphs have been revised and renamed for greater clarity and consistency.
- Additionally, we have included a new table presenting vaccination rates during the analyzed periods, to improve data interpretation and support the discussion.
We appreciate your feedback, which has contributed to improving the quality of the manuscript.
Manuscript ID: viruses-3630819
Clinical and Epidemiological Aspects of Measles in Children: The 2017–2019 and 2023–2024
Epidemics
In this manuscript, the authors present an analysis of the clinical and epidemiological
characteristics of measles infections among children in Romania during the 2017–2019 and
2023–2024 outbreaks. They notably highlight a rising incidence among older children, which
they attribute to the cumulative impact of declining immunization coverage over time.
In general, it’s better if the paper places the study within the global WHO’s goal of measles
elimination by 2030. Emphasizing the role of such studies helps show its value in supporting
this global effort.
We added following idea into the Introduction section section
The World Health Organization (WHO) has set an ambitious goal to eliminate measles in all regions by 2030, emphasizing high vaccination coverage (≥95% with two doses of measles-containing vaccine), robust surveillance systems, and rapid outbreak response as essential pillars of this strategy [1]. In this context, our study contributes valuable epidemiological data from a high-burden EU country, documenting patterns of clinical severity, and gaps in vaccine coverage during two major outbreaks over the past few years—serving as a meaningful stop along WHO’s global roadmap toward measles elimination.
Title and Keywords :
I recommend including the country name ("Romania") in both the title and the keywords to
enhance clarity and improve the discoverability of the article in databases.
This was done
Title: The title could be modified to be more relevant of the results and the impact of the
study toward the global measles elimination by 2030, the importance of vaccination
The title has been revised, and the study was conducted in a tertiary care center in Bucharest, Romania.
We added the idea into the Discussion section
"The title has been modified in accordance with the suggestions of all reviewers."
Materials and Methods:
Please specify the number of children included in each group analyzed in this section,
describe your study population and the periods of sampling. This detail is essential for
understanding the structure of the study population and for evaluating the robustness of the
statistical analysis.
We added following ideas
The study included pediatric patients aged 0 to 18 years, who were categorized into two cohorts according to their period of hospitalization: Group 1 - 1.068 cases (2017–2019) and Group 2 – 3.114 cases (2023–2024).
The data from the periods 2017-2019 and 2023-2024 were obtained from the hospital’s statistical program, Info World. We selected the period and the diagnosis of Measles infection, both as primary and secondary diagnoses, and then counted the patients.
Results:
It is advisable to avoid starting the Results section with an introductory statement. I suggest
removing the sentence on line 75 and instead beginning directly with the presentation of the
findings (example “The study showed…”).
This was done
Lines 82-83 – Sex Distribution:
The authors mention a slight predominance of male cases. Please including the p-value in the
text.
In terms of sex distribution among children, no statistically significant differences were observed between the two groups (p=0.508). We observed that the number of male children predominated compared to the number of female children (p< 0.001). The proportion of boys was 55.3% in the first group and 53.5% in the second group. (Table 1)
Graphs 1 and 2 – Inconsistency in Time Periods and Suggestion for Consolidation:
There is a discrepancy between the graph titles and their respective legends: Graph 1 refers to
the 2017–2019 outbreak in the title but is labeled as 2023–2024 in the legend, while Graph 2
presents the reverse. Please correct these inconsistencies to ensure clarity and accuracy.
Additionally, I suggest merging both graphs into a single composite figure to facilitate direct
comparison between the two outbreak periods.
Graphs 1, 2, and 3 – Presentation Style:
The current format of Graphs 1, 2, and 3 resembles PowerPoint-style visualizations, which are
less suitable for scientific publications. I recommend replacing them with standard figure
formats exported from Excel or statistical software (e.g., bar charts, line graphs) with clear
axes, legends, and labels that adhere to scientific publication standards.
Line 117 – Increase in Severity:
The authors mention an increase in the severity of cases. Could the authors clarify whether
this observation was supported by statistical analysis? Including relevant p-values or
confidence intervals would help substantiate this claim.
All tables and figures have been thoroughly reviewed and revised to ensure clarity and accuracy.
This is noted in the final paragraph of the Results section
Gastrointestinal complications (acute diarrhoea, hepatitis, dehydration syndrome, pancreatitis, stomatitis), haematological (leukopenia, anaemia, thrombocytopenia) and ophthalmological (conjunctivitis and keratoconjunctivitis) were more frequent in 2023-2024 (p < 0.00001). Neurological complications (convulsions seizures and encephalitic reaction) were more frequent in 2017-2019. (Table no. 2)
Table 2 – Title Clarification:
I suggest modifying the title of Table 2 to include the time periods and the population studied,
for improved clarity. A recommended title could be: “Measles Complications in Children
Observed During the 2017–2019 and 2023–2024 Outbreaks.” THIS WAS DONE
Table 2 – Spelling Correction – Terminology: THIS WAS DONE
Please correct the spelling of “oftalmology” to the standard scientific term ophthalmology.
General Remark on Tables – Formatting Standards: THIS WAS DONE
In scientific publications, tables should follow a clean and standardized format. I recommend
removing background colors and minimizing the use of gridlines. Tables should be presented
in plain format with only essential horizontal lines (e.g., separating the header from the data),
in line with journal guidelines and standard scientific style.
Discussion
In lines 133–134, the authors state that "the waves highlight a significant increase in the
number of pediatric measles cases during the 2023–2024 outbreak." However, they should
clarify that this significant increase was particularly observed in individuals under the age of
vaccination (12 months) and over 14 years of age.
Our study found that the last epidemic wave (2023–2024) particularly affected individuals under 1 year of age and those older than 14 years old.
This was done
We have included a new table illustrating vaccination coverage and the trend observed across the two analyzed epidemic waves.
Furthermore, in lines 135–136, the authors suggest that "this surge may be attributed to the
declining vaccination coverage, as nearly all affected children were unvaccinated." This
hypothesis is valid for the period 2020-2022, for those over 14 years old, do the authors have
any reference or evidence indicating that there was a gap in vaccination coverage during the
period when this specific age group was scheduled to be immunized?
Later in the discussion when authors speak about the MMR vaccination in Europe, we
understand that in Romania the vaccination is recommended and not mandatory, so I suggest
to be more fluent, that after giving the hypothesis of gap in immunization, authors mention
that the vaccination against Measles virus in Romania is not mandatory.
This idea was added:
In Romania, the current vaccination policy is regulated through the National Vaccination Program (NVP), coordinated by the Ministry of Health and implemented through the network of general practitioners. This policy aligns with the National Vaccination Strategy 2023–2030, which aims to increase vaccination coverage and reduce the incidence of vaccine-preventable diseases. Vaccination is not legally mandatory; the recommended vaccines are offered free of charge and are strongly encouraged as part of public health efforts.
Graph 4 does not present original results from the current study, and therefore may not be
necessary in the Discussion section. The authors are encouraged to extract the key message
from this graph and cite the original source instead. We deleted the graph 4.
Please revise all the graphs of the manuscript, it is a scientific paper not a PowerPoint
presentation. This was done

Reviewer 3 Report
Comments and Suggestions for Authors
Dear Editor, Dear Authors,
Thank You for the opportunity to review a manuscript entitled “Clinical and Epidemiological Aspects of Measles in Children: The 2017–2019 and 2023–2024 Epidemics”. The manuscript presents some descriptive data on two measles in children epidemics and in my opinion provides some evidence for surveillance and decision making. The provided information although with some potential are not free from limitations which make generalization difficult.
I suggest considering the following issues before publication:
*add please settings in the title;
*although the aim of the study is expressed in the abstract it is not provided in the main body of the manuscript, add please;
*as described under Materials and Methods the study included hospitalized cases. This is in contrast to the aim which states ‘clinical and epidemiological characteristics of paediatric measles cases during’ referring to / suggesting general population. This requires clarification;
Material and Methods:
*name the study referring to the study design types;
*page.2-line.58-60: provide setting please; population coverage should be described and a comment about representativeness and rationale for generalization on the whole Romanian population (in the Discussion paragraph);
*page.2-line.60-67: as it was a retrospective design, provide, please, what was the source of the data (medical records?); were there missingness? How have you treated with missing values?
*this was a retrospective study so how the informed consent was taken retrospectively?
*the Fisher’s exat test was used in the statistical analysis, this test however is typically used if the assumptions for the chi-squared test are not met. The use of the Fisher’s exact is a compromise as it assumes that the marginal totals are fixed, which is not always be appropriate. As it is visible in tables majority of the data fulfill the criteria for the chi-squared, so I suggest recalculating.
Results:
*Regarding data presentation in the main body of the text and in tables/graphs … there is a mixture of dots and commas which represent a decimal point, unify using English notation, please;
*p.2-l.77: as far as I understand the number of measles cases represents the total hospitalized in the Paediatric Departments of the "Prof. Dr. Matei Balș" National Institute of Infectious Diseases; relate it please, with the total number reported for Romania (comment under Discussion part)
*It would be worth knowing age of children died in both pandemics;
*p.4-l.112: the proportion of deaths varies from the calculation/percentage provided in the table 1;
*the scale on the axis x (calendar time) is not continuous; this shows the epidemic curve which is mistaken as in reality it looked differently;
*I suggest merging graph 1 & 2 to visualize better the differences;
*Graph 3: I suggest adding ‘hospitalized’ i.e. ‘Hospitalized Pediatric Measles Cases’, as it suggests the distribution for all measles cases in Romania;
Disussion:
I suggest adding in the content:
*basic or effective reproductive ratio for the epidemics evaluated
*vaccination coverage in the region presented/discussed
*discuss the results in the context of vaccination schedule in Romania and the data on measles vaccination coverage (provide proportion of vaccinated) – this should show the link between vaccination coverage and epidemics
*some information on the measles surveillance system, the strategy ‘enforcing’ data collection and its effectiveness;
Additionally:
*p.6-l.170-173: show, please, some numbers and refer to regular reports instead of Reuters;
*under the paragraph ‘Epidemiological impact on healthcare systems” discuss more about direct and indirect costs and diverted resources
*under the paragraph ‘Vaccination policies’ provide, please, the details of the measles vaccination policy in Romania;
Conclusions:
*There are well known strategies for control mentioned. I suggest adding the conclusions based on the current study, as there are none.
In my opinion the manuscript, provided if improved, is worth considering for publication.
Reviewer.
Grammar issues: like p.3-l.98: “Tables may have a An analysis of Graphs No. 1 and No. 2 demonstrates”
Author Response
Response to Reviewer 3:
Thank you for your valuable comments and suggestions. In response:
- A new author, who was not included in the initial version, has been added to the manuscript. Consequently, the order of authors has changed.
- The graphs have been revised and renamed for greater clarity and consistency.
- Additionally, we have included a new table presenting vaccination rates during the analyzed periods, to improve data interpretation and support the discussion.
We appreciate your feedback, which has contributed to improving the quality of the manuscript.
Dear Editor, Dear Authors,
Thank You for the opportunity to review a manuscript entitled “Clinical and Epidemiological Aspects of Measles in Children: The 2017–2019 and 2023–2024 Epidemics”. The manuscript presents some descriptive data on two measles in children epidemics and in my opinion provides some evidence for surveillance and decision making. The provided information although with some potential are not free from limitations which make generalization difficult.
Thank you for your review and support!
I suggest considering the following issues before publication:
*add please settings in the title; This was done
*although the aim of the study is expressed in the abstract it is not provided in the main body of the manuscript, add please;
We added the following idea
The aim of this study was to compare the clinical and epidemiological characteristics of paediatric measles cases hospitalized in our center during two major epidemic waves in Romania (2017–2019 and 2023–2024), in order to identify changes in disease presentation and outcomes over time
*as described under Materials and Methods the study included hospitalized cases. This is in contrast to the aim which states ‘clinical and epidemiological characteristics of paediatric measles cases during’ referring to / suggesting general population. This requires clarification;
We added this idea in the abstract: This study aims to highlight the clinical and epidemiological characteristics of paediatric measles cases hospitalized in Paediatric Departments of the "Prof. Dr. Matei Balș" National Institute of Infectious Diseases, Bucharest, Romania in during two distinct epidemic waves: 2017–2019 and 2023–2024.
Material and Methods:
*name the study referring to the study design types;
This was added
*page.2-line.58-60: provide setting please; population coverage should be described and a comment about representativeness and rationale for generalization on the whole Romanian population (in the Discussion paragraph);
We completed the paragraph. We conducted a retrospective descriptive study of measles cases hospitalized in the Paediatric Departments of the "Prof. Dr. Matei Balș" National Institute of Infectious Diseases, a tertiary hospital in Bucharest, capital of Romania, Southeastern Europe in the periods 2017-2019 and 2023- 2024.
"While our hospital primarily serves the population of Bucharest and Ilfov, as a tertiary care center it also manages cases referred from other regions of Romania – this is for clarification
We added into the Disscution more informations
*page.2-line.60-67: as it was a retrospective design, provide, please, what was the source of the data (medical records?); were there missingness? How have you treated with missing values?
Demographic data (age and sex) and clinical characteristics were extracted from patients' medical records.
We clarified this idea in methods. We didn’t have missing values
*this was a retrospective study so how the informed consent was taken retrospectively?
All hospitalized patients provided informed consent for participation in trials and clinical studies. A standardized consent form is used for the parents or legal guardians of minor children. Patients for whom consent was not obtained were excluded from the study.
Upon admission, all legal caregivers signed a consent form for the publication and the use of patient data for the study. The study was conducted in accordance with the Declaration of Helsinki and was approved by the Ethics Committee of the National Institute of Infectious Diseases “Prof. Dr. Matei Balș.” with identification number C03609/April 9 2025
*the Fisher’s exat test was used in the statistical analysis, this test however is typically used if the assumptions for the chi-squared test are not met. The use of the Fisher’s exact is a compromise as it assumes that the marginal totals are fixed, which is not always be appropriate. As it is visible in tables majority of the data fulfill the criteria for the chi-squared, so I suggest recalculating.
We recalculated the p-value using the chi-square test.
Results:
*Regarding data presentation in the main body of the text and in tables/graphs … there is a mixture of dots and commas which represent a decimal point, unify using English notation, please;
This was done
*p.2-l.77: as far as I understand the number of measles cases represents the total hospitalized in the Paediatric Departments of the "Prof. Dr. Matei Balș" National Institute of Infectious Diseases; relate it please, with the total number reported for Romania (comment under Discussion part)
It was wroted in Disscusion section
In Romania, the analysis of measles epidemiological data has been conducted through the extraction and centralization of available county-specific and annual reports. However, for the year 2024, national data remain partially underreported, limiting the precision of estimates regarding the total number of cases. According to reports from UNICEF and the World Health Organization (WHO), Romania reported the highest number of measles cases in the European Region for 2024, with a total of 31886 cases, accounting for approximately 87% of all reported cases in the European Union and European Economic Area
*It would be worth knowing age of children died in both pandemics;
The ages of the deceased patients have been explicitly stated in the text.
*p.4-l.112: the proportion of deaths varies from the calculation/percentage provided in the table 1;
This was done
*the scale on the axis x (calendar time) is not continuous; this shows the epidemic curve which is mistaken as in reality it looked differently;
This was done
*I suggest merging graph 1 & 2 to visualize better the differences;
*Graph 3: I suggest adding ‘hospitalized’ i.e. ‘Hospitalized Pediatric Measles Cases’, as it suggests the distribution for all measles cases in Romania;
This was done. All tables and figures have been thoroughly reviewed and revised to ensure clarity and accuracy
Disussion:
I suggest adding in the content:
*basic or effective reproductive ratio for the epidemics evaluated
Estimating the basic reproduction rate was challenging, as many patients did not disclose contact with an infected individual, which hindered the identification of the index case
*vaccination coverage in the region presented/discussed
"While our hospital primarily serves the population of Bucharest and Ilfov, as a tertiary care center it also manages cases referred from other regions of Romania
*discuss the results in the context of vaccination schedule in Romania and the data on measles vaccination coverage (provide proportion of vaccinated) – this should show the link between vaccination coverage and epidemics
*some information on the measles surveillance system, the strategy ‘enforcing’ data collection and its effectiveness;
We added this idea
In Romania, measles surveillance is conducted through a national epidemiological system coordinated by the National Institute of Public Health (INSP), in alignment with WHO and ECDC guidelines. The surveillance strategy mandates the immediate reporting of all suspected cases by healthcare providers, followed by epidemiological investigation and laboratory confirmation through serology (IgM ELISA) or molecular testing (RT-PCR). Although data collection is compulsory, the system’s effectiveness can be hindered by underreporting, delayed notifications, and limited access to diagnostic testing in certain areas. Despite these challenges, the surveillance system has proven effective in detecting outbreaks and monitoring epidemiological trends.
Additionally:
*p.6-l.170-173: show, please, some numbers and refer to regular reports instead of Reuters;
*under the paragraph ‘Epidemiological impact on healthcare systems” discuss more about direct and indirect costs and diverted resources
We added following idea
In our case, the hospitalization of moderate to severe cases due to complications has overwhelmed paediatric wards, complicated case allocation due to the high contagiousness of the virus, and—when overlapping with other epidemics such as the 2024 influenza outbreak or pertussis—has further strained hospital resources and complicated case management. Beyond the clinical burden, the outbreak has led to significant direct costs, including increased expenditures for hospitalization, intensive care, diagnostics, supportive therapies. Indirect costs have also risen, caused by caregiver absenteeism, school closures, and long-term health impacts in affected children.
*under the paragraph ‘Vaccination policies’ provide, please, the details of the measles vaccination policy in Romania;
We added In our country, the current vaccination policy is regulated through the National Vaccination Program (NVP), coordinated by the Ministry of Health and implemented through the network of family doctors. This policy aligns with the National Vaccination Strategy 2023–2030, which aims to increase vaccination coverage and reduce the incidence of vaccine-preventable diseases. Vaccination is not legally mandatory; the recommended vaccines are offered free of charge and are strongly encouraged as part of public health efforts.
Conclusions:
*There are well known strategies for control mentioned. I suggest adding the conclusions based on the current study, as there are none.
The conclusions have been reformulated to reflect the findings of our study and now also incorporate strategies to improve vaccination coverage.
The study further emphasizes the necessity of both specific interventions—such as the timely administration of measles immunoglobulin to high-risk children—and non-specific preventive strategies aimed at reducing transmission. Overall, our data reinforce the urgent need to address declining vaccination rates and implement effective public health strategies to control measles transmission.
In my opinion the manuscript, provided if improved, is worth considering for publication.
Reviewer.
Grammar issues: like p.3-l.98: “Tables may have a An analysis of Graphs No. 1 and No. 2 demonstrates”
This was done

Round 2
Reviewer 3 Report
Comments and Suggestions for Authors
Dear Authors & Editor,
The submitted revision of the manuscript includes several improvements and addresses majority of the issues mentioned in my first review. There are some minor editorial elements which are worth being corrected before final publication, as:
*the decimal points in the tables are printed as commas; correct into dots, please
*Graph 1 has non-linear time on axis x (the same distances between Oct->Dec /1 month, Dec->Feb / 2 months; Feb->May / 3 months ) which does not provide a proper description of the epidemic curve;
*Graph 3: a mixture of dots and commas decimals;
Overall, the scientific soundness of the manuscript and its clarity have been improved. I suggest accepting the paper for publication. Addressing one of the major world public health issue the presented paper leads to the promise to be highly cited.
Reviewer
Author Response
Response to Review Round 2:
Thank you for your valuable comments and suggestions. In response
The submitted revision of the manuscript includes several improvements and addresses majority of the issues mentioned in my first review. There are some minor editorial elements which are worth being corrected before final publication, as:
*the decimal points in the tables are printed as commas; correct into dots, please.
This was corrected.
*Graph 1 has non-linear time on axis x (the same distances between Oct->Dec /1 month, Dec->Feb / 2 months; Feb->May / 3 months ) which does not provide a proper description of the epidemic curve;
Graph 1 and 2 have been redone
*Graph 3: a mixture of dots and commas decimals;
This was corrected.
Overall, the scientific soundness of the manuscript and its clarity have been improved. I suggest accepting the paper for publication. Addressing one of the major world public health issue the presented paper leads to the promise to be highly cited.
Thank you very much for the suggestions, I tried to modify it according to your requests and I hope it will be a useful work
